# Indole Alkaloids from Psychoactive Mushrooms: Chemical and Pharmacological Potential as Psychotherapeutic Agents

**DOI:** 10.3390/biomedicines11020461

**Published:** 2023-02-05

**Authors:** Erika Plazas, Nicoletta Faraone

**Affiliations:** Department of Chemistry, Acadia University, Wolfville, NS B4P 2R6, Canada

**Keywords:** psychedelics, tryptamines, β-carbolines, ergot alkaloids, biosynthesis, pharmacology, neuropsychiatric disorders, toxicology, clinical trials

## Abstract

Neuropsychiatric diseases such as depression, anxiety, and post-traumatic stress represent a substantial long-term challenge for the global health systems because of their rising prevalence, uncertain neuropathology, and lack of effective pharmacological treatments. The approved existing studies constitute a piece of strong evidence whereby psychiatric drugs have shown to have unpleasant side effects and reduction of sustained tolerability, impacting patients’ quality of life. Thus, the implementation of innovative strategies and alternative sources of bioactive molecules for the search for neuropsychiatric agents are required to guarantee the success of more effective drug candidates. Psychotherapeutic use of indole alkaloids derived from magic mushrooms has shown great interest and potential as an alternative to the synthetic drugs currently used on the market. The focus on indole alkaloids is linked to their rich history, their use as pharmaceuticals, and their broad range of biological properties, collectively underscoring the indole heterocycle as significant in drug discovery. In this review, we aim to report the physicochemical and pharmacological characteristics of indole alkaloids, particularly those derived from magic mushrooms, highlighting the promising application of such active ingredients as safe and effective therapeutic agents for the treatment of neuropsychiatric disorders.

## 1. Introduction

Neuropsychiatric disorders (NSDs) comprise a group of complex mental conditions with varied epidemiology and uncertain pathophysiology, mostly characterized by behavioral and mood alterations associated with brain malfunction [1]. Major NSDs include depression, anxiety, and neurotic and psychotic conditions, such as obsessive-compulsive disorders, depressive disorders, post-traumatic stress, bipolarity, and schizophrenia, among others [2]. Mental disorders represent a substantial long-term challenge for the global health system, because of their rising prevalence, uncertain neuropathology, and lack of effective pharmacological treatments [3,4]. A recent analysis of 12 mental disorders in 204 countries around the world revealed a growing burden tendency linked to the prevalence and disability rates of these conditions. In fact, this study estimated an increase of around 48% in cases of mental illnesses between 1990 and 2019, with anxiety and depressive disorders being the most prevalent with nearly 580 million cases in 2019 [5]. Additionally, given the chronic characteristics of some NSDs, years lived with disability (YLDs) turned out to the marker with a higher contribution to the global burden (125 million in 2019), as it implies a deterioration in patient’s quality of life, which entails severe social and economic impacts [5].

Furthermore, as if the above statistics were not disquieting enough, an exacerbated increase in cases of depressive and anxiety disorders is expected because of the COVID-19 pandemic. Different psychiatric and epidemiological studies have been warning social and economic circumstances faced during and after the pandemic might be leading to irreversible effects on the mental health of children, youth, and adults [6,7,8]. Furthermore, recent reports have suggested the potential impact of acute SARS-CoV-2 infection on the brain and central nervous system as a side effect of the hyperinflammatory cascade, which could lead to cognitive and neurological sequela in COVID-19 patients [7,9]. Preliminary evidence showed that about 23% of 40,000 patients presented neuropsychiatric symptoms, including anxiety, stress, and psychosis; however, more in-depth research is required to determine the tangible magnitude of neurological impacts in recovered COVID-19 patients [7]. For instance, all the above evidence indicates ultimately that the effects of the pandemic might entail an even greater challenge in the burden of neuropsychiatric disorders and global public health. Therefore, the search and implementation of strategies focused on the prevention and efficient treatment of mental disorders are an essential need in current and future society.

Research and development (R&D) of safe and effective drugs could be a keystone in the management and control of neuropsychiatric disorders burden. Nevertheless, the landscape in R&D of new psychotherapeutic agents appears bleak given the modest success rates of candidates in clinical phases, which have been mostly associated with the poor understanding of the neuropathophysiology of NSD hindering either target selection and molecular mechanism elucidation [10]. Moreover, some long-term antidepressant drugs have proven loss of efficacy, reduction of sustained tolerability, and unpleasant side effects [11,12]. Indeed, recent analyses have questioned their sufficiency and impact on the patient’s quality of life, revealing the immediate need for therapeutic alternatives for NSD [3,13]. Thus, different strategies have been explored to minimize the failure of drug candidates, advancing the elucidation of their mechanism action, abating the effectiveness gap, as well as broadening the sources of new successful therapeutic agents [1,10,14,15]. In recent years, the revival of natural products on the drug development landscape has been gaining assets to both the scientific community and pharmaceutical industry, especially in the field of neuropsychiatric disorders and complex diseases [16,17]. Largely, this re-emergence has been driven by breakthrough research of natural psychedelics for the treatment of mental illnesses such as anxiety and depression. Undoubtedly, indole-type alkaloids, such as psilocybin, harmine (i.e., ayahuasca alkaloids), and dimethyltryptamine (DMT) among others, have become the epicenter of this recent psychedelic-medicinal wave, as it is discernible in the rising number of publications, research programs, and clinical trials involving these metabolites [16,18,19]. In fact, mushroom-derived indole alkaloids have demonstrated an extraordinary therapeutic potential for neuropsychiatric conditions [20,21]. Unfortunately, the chemical and pharmacological studies of these alkaloids have been overshadowed for decades due to their illegal and psychotropic connotation, hampering the discovery of potential bioactive natural alkaloids derived from magic fungi.

This compilation of the literature provides a comprehensive review of the physicochemical and pharmacological characteristics of indole alkaloids, particularly those derived from magic mushrooms, and aims to establish a reference guideline for researchers interested in the search, rational design, and development of psychotherapeutic agents inspired by naturally occurring alkaloids. For this purpose, this review presents an introductory chronological overview of the therapeutical uses of magic mushrooms, emphasizing the ethnopharmacology and historical inflection points that restrain the scientific research. Then, the distribution and chemotaxonomic relevance of indole-type alkaloids as linked to their evolutionary role in the fungus kingdom will be displayed. Subsequently, the chemical features will be addressed, targeting the biosynthesis pathways, extraction, and analytical methods of selected examples with commercial interest. The final part of the review will spotlight the psychopharmacological potential of fungi-derived indole alkaloids, through the discussion of different representative cases, targets, and proposed mechanisms, plus pharmacokinetics and toxicology traits, ending with a synopsis of ongoing clinical trials ( https://www.clinicaltrials.gov/ accessed on 10 October 2022) with some of those indole alkaloids.

## 2. Ethnomycology and Historical Overview of Psychoactive Mushrooms around the World

Psychoactive fungi comprise a diverse group of basidiomycetes characterized by the ability of inducing different neurotropic effects after ingestion, including mental and mood alterations, as well as hallucinations [22]. Historically, humans have used natural products to supply daily requirements, as a source of materials, food, cosmetics, and medical remedies. Additionally, extensive evidence has proven the use of natural psychedelics derived from plants and fungi for mental healing, as well as for spiritual-ritual and recreational purposes since ancient times [22,23,24]. As such, hallucinogenic mushrooms have been widely used in the folk medicine of different indigenous communities around the world for the treatment of physical and mental conditions, as well as to induce altered consciousness stages for magic rituals and sacred ceremonies, particularly in American and Oceania aboriginal cultures [25]. In fact, the use of psilocybin-containing fungi, also known as magic mushrooms, constitute a ceremonial shamanic tradition of Mesoamerican indigenous cultures. Mayan and Aztec civilizations are an unprecedented example in terms of heritage usage of magic mushrooms, and this is reported in mushroom-shaped sculptures that date back to 500–200 B.C symbolizing the significance of magic fungi in Mayan cultures [25]. Moreover, according to anthropological studies, there is evidence of the magico-ritual practices with at least 15 different *Psilocybe* species by the Aztecs throughout the Mexican territory [22,26]. For instance, *P. caerulescens* and *P. mexicana* (Table 1), popularly called as *Teotlaquilnanácatl,* meaning “sacred mushroom that paints”, were reported in the 16th century in Magliabechiano Codex and are still currently used by *Nahuatl* shamans as an entheogen to achieve transcendental states and advanced forms of consciousness in sacred ceremonies [26].

Other representative examples of psychoactive mushrooms and their worldwide ethnomycological importance are presented in Table 1. *Amanita muscaria,* also known as “fly agaric”, is one of the most recognized hallucinogenic mushrooms, known not only for its colorful appearance but also for its psychotropic use and poisonous properties [22]. The *pantherina* poisoning syndrome caused by *Amanita* ingestion is rarely fatal and atropine-like, and its symptoms include dilated pupils (e.g., mydriasis), space distortion, and visual and auditory aesthesia. After a few hours of poisoning gastrointestinal discomfort, central nervous system dysfunctions might be presented, and some studies suggest that systematic consumption of *Amanita* mushrooms might induce brain lesions [27]. Despite their well-known history of toxic properties, *A. muscaria* has played a significant role in the ethnopharmacology of different cultures worldwide, standing out for its mystical and sacred role. In fact, hundreds of European myths described the use of fly agaric in elixirs and magic potions prepared by witches and wizards. Additionally, in ancient Greece, this mushroom, known as the “food of the gods”, was the ritual of Orphic and Eleusinian mysteries, and it is *“believed to be part of the soma”* in ancient Vedic-Aryan tribes [25]. Moreover, there is extensive controversy regarding the edible and medicinal uses of *Amanita*, and according to the ancestral knowledge of some North American indigenous tribes (Table 1), *A. muscaria* (i.e., *Miskwedo*) can be used to treat mental ailments, generating states of relaxation and pleasure, without causing side effects [28]. Nevertheless, at present, there is still lack of scientific evidence to support the traditional knowledge and medicinal properties of *Amanita* mushrooms.

Different authors have described the consumption of psychoactive *Gymnopilus purpuratus* mushrooms by indigenous communities of the central Amazon, including countries such as Brazil, Ecuador, Peru, and Colombia for healing and spiritual purposes (Table 1). In fact, the *Gymnopilus* species are believed to be used in folk medicine by *Paumarí* Indians in Brazil, whereas other species could be mixed and smoked as a snuff to cure mental and soul afflictions [21]. However, it is worth pointing out that there is a boundless disagreement around the unambiguous classification of magic mushrooms used in shamanic rituals by indigenous communities before the pre-Columbian era because of the absence of taxonomic tools; therefore, studies of characterization as well as the validation of ethnomycological uses are still required.

Some magic mushrooms, such as those belonging to the *Inocybe* genus, do not have reports of traditional uses and instead were discovered by accidental hallucinogenic poisonings. For example, *I. aeruginascens* (Table 1) was described as psychoactive in the 1980s from unintentional intoxication in Germany and Hungary because of its similarity to the edible mushroom *Marasmius oreades.* Despite the unintentional consumption of *I. aeruginascens,* the user reported an extremely pleasant experience, described as a “good trip”. This feature caught the attention of the German chemist Jochen Gartz, who discovered a new alkaloid structurally related to those found in other species of magic mushrooms, although in turn, has different pharmacological properties [30].

According to the worldwide ethnomycology, a transversal feature of psychedelic mushrooms is their holistic use in the management of mental healing, as inducers of calm, relaxation, and happiness states (Table 1). Perhaps this ethnopharmacological knowledge also played a key role in prompting the early research in the field of mental health and emerging interest in the therapeutic potential of psychedelics in the 1950s. Nevertheless, it is known that the era of psychedelics in psychiatry was sparked by the synthesis of lysergic acid diethylamide (LSD) in 1938 and the subsequent serendipity discovery of its psychedelic effects in 1943, both by Swiss chemist Albert Hofmann [33].

Between 1943 and 1960 there was a burgeoning development in natural-derived psychedelics research, as well as a growing interest in the use of these for the treatment of mental illnesses (Figure 1). Within this historical development, it is worth pointing out the official introduction of the currently used term “psychedelic” (1956) to refer those chemical substances that are able to alter the perception of space–time, inducing altered states of consciousness, as well as the term ‘magic mushrooms’ (1957) to evoke a group of fungi containing hallucinogenic compounds [34]. This was followed by the identification of the first psychoactive mushroom-derived alkaloid, with the isolation of the psilocybin from fungi of *Psilocybe* genus in 1958 (Figure 1). Certainly, psilocybin discovery opened the door to a new uncharted area of psychedelic natural products, promoting the chemical research of these compounds. Indeed, different structurally related psilocybin alkaloids were isolated and identified between 1958 and 1965, and all of them were further classified as tryptamine derivatives by Hoffer and Osmond [35]. On the pharmacological side, in 1960, Dr. Timothy Leary started the controversial “Harvard psilocybin project” to evaluate its effects in human behavior through the administration in voluntary students. This study triggered a heated ethical debate around the use of hallucinogens, which years later led the prohibition of LSD and psilocybin, as well as scientific research related to the use of psychedelics in psychiatry (Figure 1). Hence, the legislative actions adopted by the governments truncated the progress accomplished in the elucidation of the therapeutical potential of natural psychedelics, and most of the research in this field conducted during the 70s and 80s was kept hidden [34].

Despite the diplomatic conflict and after decades of latency, the research in psychedelics entered a stage of a renaissance in the early 21st century as a result of the persistent interest in their pharmacological potential by some medical-psychiatric scientists and private research institutions [20,21]. In this context, one of the flagship projects that has driven this renewal is the magic mushrooms study at the Johns Hopkins Center, showing the high safety and efficiency of psilocybin for challenging mental disorders [36]. Furthermore, there are remarkable achievements in the clinical trials with 3,4-methylenedioxymethamphetamine (MDMA) for the management of alcoholism, social anxiety, and post-traumatic stress disorder (PTSD) [37]. Recently, after a tangled pathway, the Food and Drug Administration (FDA) not only began to support clinical studies with MDMA and psilocybin, but also assigned them the category of promising therapies for the treatment of depression [34]. Currently, psychedelics research is at Its peak, and the last two years has witnessed an exponential growth in the number of studies, clinical trials, publications, and licenses to research in this area [38,39,40,41].

## 3. Distribution of Psychoactive Mushrooms and Chemotaxonomic Relevance of Indole Alkaloids

According to Guzman et al. [42], psychoactive mushrooms can be divided into four main classes based on the chemical compounds associated with their hallucinogenic actions (Table 2). The first group comprises different basidiomycetes of the *Psilocybe*, *Gymnopilus*, *Panaeolus*, *Hypholoma*, *Pluteus*, *Inocybe*, and *Conocybe* genus, commonly denominated as magic mushrooms, which are characterized by containing tryptamine or related indole alkaloids, such as psilocybin (4-phosphoryloxy-*N*,*N*-dimethyltryptamine) and psilocin (4-hydroxy-*N*,*N*-dimethyltryptamine) [22]. Given the structural relationship between these tryptamine derivatives and the neurotransmitter serotonin, magic mushrooms are also recognized for exerting their psychotropic action by interacting with serotonin (5-HT2A) receptors [25].

Mushrooms characterized by the presence of psychoactive markers with amino acid isoxazole scaffolds are classified in the second group (Table 2). Essentially, these are fungi of the *Amanita* genus, which produce compounds such as muscimol (5-(Aminomethyl)-1,2-oxazol-3(2H)-one) and ibotenic acid ((*S*)-2-Amino-2-(3-hydroxyisoxazol-5-yl)acetic acid) [22]. *Amanita* biomarkers act as agonists on GABA (gamma-aminobutyric acid) receptors, inducing sedative-hypnotic effects, synesthesia, and other altered-mind states [27]. On the other hand, the third group is composed of ascomycetes of the *Claviceps* and *Cordyceps* genus, which are endoparasitoid fungi in plants and insects, respectively. The chemo-markers of this group are a type of indole alkaloids that share the ergoline scaffold (Table 2), and consequently are known as ergoline derivatives or ergot alkaloids [22]. Given the structural complexity of ergot alkaloids, they have been divided into two subgroups, the amide-type such as lysergic acid, and the ergopeptine-type for those linked to amino acid units such as ergotamine. In addition, their distinctive neurotransmitter-like structural traits provide them multi-target action by modulating serotonin, dopamine, and adrenergic receptors [43]. Finally, in the fourth group, Guzman suggested to gather basidiomycetes of the *Russula*, *Boletus*, and *Heimiella* genera, as well as other gasteroid mushrooms, which are recognized as sacred in different tribes worldwide but lack accurate chemical identification of their psychoactive compounds. In this context, chemical and pharmacological studies featuring these basidiomycetes are still required to have their psychoactive markers identified.

The diverse group of psychotropic mushrooms comprises more than 200 species belonging mainly to the Basidiomycota phylum (with approx. 20 genera) and a few to Ascomycetes (*Claviceps* and *Cordyceps*) and are distributed from the south of Chile to Alaska and Siberia in the northern hemisphere, as well as from the west coast of America to Europe, Asia, and Oceania (Figure 2). Furthermore, hallucinogenic mushrooms have been found in a wide diversity of ecosystems from sea level to high mountain regions up to 4000 m altitude [42]. The largest distribution of psychoactive mushrooms with more than 15 species is found in North, Central, and South America, West Europe, India, Japan, and Oceania (Figure 2). Other South American and East-European countries, as well as China and Russia, among others, present a medium diversity of psychedelic mushrooms with about 5 to 15 species described. Countries with lower diversity are located in Central and South America, Caribe, Africa, and the Middle East, with less than five species documented [22,42]. *Psilocybe* is the most representative psychotropic genus owing to its extensive distribution and diversity, accounting around 50% of hallucinogenic mushrooms distributed in subtropical, mesophytic, or humid regions [42]. The largest number of *Psilocybe* mushrooms is found in South America and Mexico (59 species), North America (18 species), as well as in the Caribbean region and Oceania. Mexico represents the richest biodiversity in terms of magic psilocybes with 42 species of the 116 worldwide reported. By contrast, it is worth pointing out that even though Europe and other countries of the northern hemisphere present lower diversification (number of species), they are characterized by having a high concentration (extension area) of its specimens [22,42].

Regarding the distribution and chemotaxonomy of magic mushrooms, it is crucial to highlight that most of the reports date back to the 1980s and 1990s; thus, it is essential to undertake new taxonomic studies using molecular tools and other cutting-edge technologies to improve, update, and strengthen the existing data.

## 4. Chemical Features of Mushroom-Derived Indole Alkaloids

### 4.1. Diversity and Biosynthesis of Mushroom-Derived Indole Alkaloids

Alkaloids are one of the largest classes of specialized metabolites structurally characterized by the nitrogen-containing scaffold, which are widely distributed in plants, and sporadically in fungi and animals. Remarkably, phylogenetic studies have suggested that distribution patterns and structural diversification of alkaloids are unique traits related to their role in chemical defense as an adaptive evolutionary strategy [44]. In fact, those metabolites have the greatest structural diversity in nature, accounting more than 27,000 molecules organized in diverse classes, ranging from simple to elaborated scaffolds. Despite their distinctive vast chemical landscape, alkaloids, similar to any other natural products, are biosynthesized from small building blocks, which in this case are mainly amino acids [45].

The fungal kingdom has a lower diversity of alkaloid classes compared to plants; in fact, the Basidiomycota phylum produces primarily indole-type alkaloids [46]. These alkaloids encompassing simple indoles, tryptamines, β-carbolines, terpene-indoles, and carbazoles share the indole core consisting of a pyrrole fused to a benzene ring (Figure 3A). In Basidiomycetes, specifically in psychoactive mushrooms, the most representative indole alkaloids are indolamines, which can be subdivided into two subgroups, tryptamines and ergolines (Figure 3A). Moreover, other indole-alkaloids such as β-carbolines and indole-terpenes have also been reported, either in less abundance or with restricted distribution within a genera or species [25,35,47].

Biosynthetically, indole alkaloids are derived from the aromatic amino acid L-tryptophan, which in turn is obtained from the shikimate route [45,48]. In addition, depending on the incorporation of dimethylallyl pyrophosphate (DMAPP) units in the biosynthetic pathway, indole alkaloids can be classified into two groups: non-isoprenoids (i.e., tryptamines, β-carbolines, carbazoles) and isoprenoids (i.e., ergot, terpene-indoles). As exemplified in Figure 4, in the biosynthesis of indole-derivatives, the precursor is transformed through a series of multistep enzymatic reactions, involving mainly decarboxylation, methylation, and hydroxylation conversions [49]. In the last decade, significant progress in the elucidation of fungal biosynthetic pathways has been achieved, mainly driven by the implementation of molecular and genetic tools. Interestingly, it has been found that fungal metabolic pathways related to ecological traits are usually encoded in “gene clusters”, which can be highly specific to particular taxonomic groups but may also occur in unrelated or distant taxa, as a result of a highly dynamic evolution involving horizontal gene transfer, fast degeneration, and renovation of metabolic routes. In this context, metabolic pathways and their gene clusters are a key challenge in the Fungal kingdom study to understand the relationship between the evolutionary, ecological, chemical, and pharmacological role of fungal-derived metabolites. Moreover, it is worth emphasizing that there is still a huge gap in the elucidation of biosynthetic pathways of fungal-derived metabolites. Thus far, most of the biosynthetic pathways for those specialized metabolites, including indole alkaloids, are theoretical proposals based on plant-routes, which require genomic and transcriptomic studies to be supported and corroborated.

Certainly, psilocybin is the mushroom-derived indole alkaloid with one of the highest characterized biosynthetic pathways, with gene clusters (Psi) and their enzymes identified from *Psilocybe cyanescens* and *Psilocybe cubensis*. Four key enzymes are involved in the obtention of psilocybin (Figure 3B): L-tryptophan decarboxylase (PsiD), P450 monooxygenase (PsiH), a kinase (PsiK), and S-adenosyl-l-methionine (SAM)-dependent N-methyltransferase (PsiM). Interestingly, biosynthetic studies have shown that some of these enzymes are selective to the substrates leading to almost direct obtention of psilocybin [49]. For example, the monooxygenase PsiH catalyzes the oxidation of tryptamine yielding 4-hydroxytriptamine; however, it is not able to oxidase L-tryptophan. Likewise, the kinase PsiK involved in the phosphorylation at C4 has shown preference for decarboxylated substrates such as 4-hydroxytriptamine, whereas the PsiM methyltransferase is highly selective to phosphorylated substrates such as norbaeocystin (Figure 3B). In this context, the biochemical machinery of *Psilocybe* mushrooms can be used to address alternative strategies for in vitro production of psilocybin, psilocin, and analogs.

β-carbolines such as harman and harmine are also biosynthetically derived from tryptamine (Figure 3B) and have been detected in different psychedelic mushrooms extracts, including the *Psilocybe* species [50]. In addition, some in vitro studies tested the incorporation of the L-tryptophan stable-isotope label (^13^C_11_-L-tryptophan) in *Psylocibe mexicana* mycelium to validate the biosynthesis of β-carbolines in magic mushrooms [50]. Even though β-carbolines are recognized bioactive indole alkaloids not only reported in plants but also in different Basidiomycetes, current knowledge of the genome and proteome involved in their biosynthesis remains uncertain and poorly studied. Thus, biosynthetic pathways for fungal-derived β-carbolines as shown in the Figure 3B are essentially hypothetical, opening up the possibility of innovative research areas on psychedelic mushrooms.

On the other hand, terpen-indole and ergot alkaloids (i.e., isoprenoid-indole class) present higher structural complexity compared to other indole derivatives; consequently, their biosynthetic pathways usually further require sophisticated biochemical batteries [51,52]. In addition, their pharmacological relevance has been the driving force in the elucidation of gene clusters and molecular mechanisms implicated in fungal-biosynthetic routes [51,53]. Thus, through different studies using feeding isotope-label precursors in in vitro fungi cultures, some steps have been established involving synthesis ergot alkaloids such as agroclavine and D-lisergic acid (Figure 3B). The first step of ergot alkaloid biosynthesis involves the prenylation of L-tryptophan with dimethylallyl pyrophosphate (DMAPP) to obtain the alkylated intermediate (DMAT), which sparks the subsequent N-methylation, generating the N-Me-DMAT. Afterward, a series of oxidation-reduction enzymes catalyse the intramolecular cyclization of N-Me-DMAT forming the third six-carbon. The chanoclavine-I-aldehyde (Figure 3B) is a transverse precursor of all ergot alkaloids, and this compound undergoes a second intramolecular cyclization yielding the distinctive tetracyclic ergot-type scaffold [52]. Likewise, chanoclavine-I-aldehyde is considered a branch point because subsequent enzymatic oxidations and derivatizations allow for the synthesis of other characteristic psychedelic ergot alkaloids such as D-lysergic acid, ergotamine, and fumigaclavines, among others. Ultimately, the recent breakthroughs in the biosynthetic pathways of ergot alkaloids have allowed for the implementation of biotechnological production of some molecules with high demand at the pharmacological level. However, more research in metabolic engineering and synthetic biology is still required to improve the stability, yield, and cost-benefit, as well as alternative technologies in the efficient largescale production of bioactive alkaloids.

### 4.2. Chemical Characteristics, Extraction Methods, and Alternative Production Techniques

The indole backbone (2,3-benzopyrrole) confers to this group of alkaloids distinctive electronic and structural traits, which are directly linked to their outstanding pharmacological properties. The benzopyrrole heterocycle is characterized by having a π-excessive structure (e.g., ten π-electrons) and weak basicity since the electrons on the nitrogen atom are not accessible for protonation. In fact, indole derivatives can be protonated with strong acids at the C-3 position instead N-1, producing an intermediate thermodynamically more stable. Consequently, the C-3 enamine-type reactivity makes these indole alkaloids more likely to degrade under acidic conditions, and tryptamines are more unstable at lower pHs than other non-isoprenoids indoles [54]. Nevertheless, some structural features can increase the long-term stability of indole alkaloids, such as substitution degree, type of substituents, *N*-methylation for tryptamines, and more rigid scaffolds with a higher number of cycles (like β-carbolines and carbazoles). The characteristics of reactivity, stability, and solubility are critical to avoid the degradation of some indole alkaloids during the extraction and storage phases as exemplified later.

Tryptamines are simple indole-derived alkaloids formed by a benzopyrrole with an ethylamine-type side chain at C-3, and substituents at C-4 and C-5 positions on the indole core. Because of their close structural relationship with serotonin (5-hydroxytryptamine 5-HT), some tryptamines present psychotropic activity by modulating serotonin receptors [25]. Thus far, around 80 mushrooms species have been reported to produce non-hallucinogenic and hallucinogenic tryptamines (selected examples presented in Table 2). Mushrooms containing tryptamines belong to 7 families and 12 genera, and within the *Psilocybe* genus stand out because of the highest distribution and abundance of these alkaloids [35]. Certainly, psilocybin (*O*-phosphoryl-4-hydroxy-*N*,*N*-dimethyltryptamine) and psilocin (*N*,*N*-dimethyltryptamine) are the most widely studied tryptamines found in magic mushrooms. Psilocin is the active metabolite, responsible for the hallucinogenic effects whereas psilocybin can act as a prodrug and it is converted into the active form after enzymatic dephosphorylation in vivo [55]. Both psilocybin and psilocin are chemically unstable; however, it is believed that the phosphate group on psilocybin also improves the stability of the active metabolite, preventing degradation in the mushrooms, biomass, and extracts [20,56]. Some procedures and solvents favor the phosphate ester hydrolyzation in psilocybin, and the subsequent enzymatic oxidation of psilocin leading the develop of the characteristic blue colorations in the mushroom extracts by formation of quinoid oligomers [57,58]. In this context, the key steps to guarantee their stability during extraction are enzyme inactivation and dehydration. For example, recent studies have demonstrated a high rate of tryptamines degradation in the storage fresh biomass, even keeping it at −80 °C [56]. On the other hand, methanol is reported as the most suitable solvent for the extraction of tryptamines from fungal biomass either using conventional or advanced methods, such as maceration or ultrasonic assisted extraction (UAE) [59,60,61]. However, recent studies have shown that acidification of methanol with 0.5% (*v*/*v*) acetic acid improves tryptamines extraction yields. Additionally, the effect of subsequent extractions and the homogenization techniques were investigated using the dried fungal biomass of *P. cubensis,* finding that two subsequent extractions and the use of a vortex increase the analytes concentrations in extracts [56]. In addition to psilocybin and psilocin, other tryptamines such as baeocystin, norbaeocystin, aeruginasin, and bufoteninee have been detected in lower concentrations and in certain species of psychedelic mushrooms (Table 3). These alkaloids are structural analogs of psilocybin, and have also demonstrated psychoactive properties; however, they have been less investigated on chemical and pharmacological levels [20].

Baeocystin (4-phosphoryloxy-*N*-methyltryptamine) and norbaeocystin (4-phosphoryloxytryptamine) are two psilocybin analogs originally isolated from *Psilocybe baeocystis*, subsequently detected in various *Psilocybe* species and also other psychoactive mushrooms belonging to the *Conocybe*, *Inocybe*, and *Panaeolus* genera [35]. Structurally, these tryptamines differ from psilocybin in the *N*-methylation pattern at the side chain nitrogen (Table 3). Chemically, baeocystin and norbaeocystin are similar to psilocybin in terms of polarity, reactivity, and stability, and are also unstable in the enzymatic oxidative degradation in fresh fungal biomass [20]. Likewise, aeruginascin (Table 3) is the trimethyl quaternary ammonium analogue of psilocybin. This tryptamine was isolated for the first time from the hallucinogenic mushroom *Inocybe aeruginascens* and later detected in some *Psilocybe* species such as *P. azurescens*, *P. cyanescens*, and *P. cubensis* [56,68]. This alkaloid is found mostly in the zwitterionic form (at pH 3–6), presenting higher polarity than psilocybin. Aeruginascin was extracted from dried mushrooms using acidified methanol with formic or acetic acid, or using hydroalcoholic mixtures such as methanol–water 80:20 [56,62]. Moreover, this quaternary ammonium salt presents higher stability than other phosphorylated tryptamines, showing slow degradation in dry biomass [62]. The procedures and extraction methodologies described so far as the most suitable in the isolation of psilocybin, psilocin, baeocystin, norbaeocystin, and aeruginascin from magic fungi are summarized in Table 3.

Bufotenine is psychoactive tryptamine initially obtained from amphibians of the *Bufo* genus, commonly considered a toad toxin (Table 3). However, its occurrence has also been confirmed in other organisms such as plants, mainly in Fabaceae (*Anadenanthera* genus), as well as in hallucinogenic fungi [55,69]. In mushrooms, this alkaloid has a restricted distribution in specific sections of the *Amanita* genus, and so far, have been reported in *A. citrina*, *A. porphyria*, and *A. rubescens*. Hence, bufotenine may play an important chemotaxonomic role in the differentiation of the *Amanita* species, considering that species with a higher concentration of this alkaloid are more toxic [35]. Chemically, bufotenine is an N-methylated analog of serotonin and a positional isomer of psilocin with the hydroxy substituent at the C5, contrary to the more common tryptamines in magic mushrooms characterized by the substituent at C4. This simple structural variation confers to bufotenine distinctive pharmacological and stability characteristics, which are different with respect to other hallucinogenic tryptamines [20]. In fact, this alkaloid is considered more stable than polymerization, allowing for classic alkaloid extraction procedures such as acid-base and Soxhlet with methanol [63,64].

On the other hand, the β-carboline alkaloids have lately been detected in magic mushrooms, even though these alkaloids are commonly found in other fungi where no previous reports of their occurrence in *Psilocybe* species have been reported. However, through a HPLC-MS analysis of different *Psilocybe* extracts, the presence and in vitro production of β-carbolines such as harman, harmol, harmine, and cordysinins (Table 3) was confirmed in *P. Mexicana*, *P. cubensis,* and *P. semilanceata*. Mushroom-derived β-carbolines are chemically characterized by the presence of a fully aromatic scaffold composed by a pyridine heterocycle fused to the indole core (pyrido[3,4-b]indole). This tricyclic planar structure is highly stable and presents photochemical properties, such as fluorescence. Moreover, because of the presence of two nitrogen atoms with individual acid-base characteristics (i.e., pyrrole and pyridine), β-carbolines can exist in different pH-dependent forms. All the above chemical traits provide to these alkaloids noteworthy pharmacological activity that will be discussed later [70]. Simple carbolines such as harman, harmol, and harmine can be obtained from dried fungal biomass using acid-base extractions and organic solvents such as dichloromethane [50]. Moreover, more polar carbolines such as cordysinins, brunnenins, and β-carboline-1-propanoic acid (Table 2) can be extracted with methanol or aqueous methanol using conventional maceration or UAE [65,71]. Additionally, it is important to highlight that carbolines have also been widely reported in edible and medicinal basidiomycetes. The edible mushroom-derived carbolines have complex structures with a unit of alkaloid linked to amino acid residues [55,71]. This class of alkaloids has been less explored than tryptamines and simple carbolines and could constitute an interesting source of bioactive molecules.

Isoprenoid indole alkaloids (IiAs) are recognized by a great structural diversity, encompassing numerous scaffolds from tetracyclic alkaloids to highly complex dimers. This structural heterogeneity derives from their mixed biosynthetic pathway through the union of two basic building blocks and endow them with privileged chemical and multi-pharmacological features [55]. Ergot or hemiterpenoid comprise one of the largest groups of fungal-derived nitrogenous metabolites, with more than 80 compounds mainly isolated from species of the *Claviceps* genus. This class of indole alkaloids share a common tetracyclic system (Table 3) biosynthetically formed by tryptophan and DMAPP condensation [43]. The non-fully aromatic ergot-scaffold is well-known by containing various chiral centers leading to different stereoisomers. Thus, it has been reported that fungal-derived ergot alkaloids are usually found as mixtures of isomers because of spontaneous epimerization during purification processes [67]. Additionally, according to the substitution patterns on the D ring, they are sub-classified into two main groups: clavines such as lysergol, as well as lysergic acid derivatives such as paspalic acid and ergonovine, respectively (Table 3).

For the extraction of ergot alkaloids from dry and ground material, different solvent systems, conditions, and matrix effects have been investigated in order to optimize the stability and efficiency of this process [67]. These studies have shown the use of organic solvents (such as dichloromethane, chloroform, or acetonitrile) at alkaline pH improves extraction yield in comparison with hydroalcoholic acidified mixtures, since acid conditions favour the isomerization processes affecting recovery yields [67]. The study of ergot alkaloids has been promoted not exactly by their remarkable pharmacological and psychoactive properties, rather because they are classified as toxin contaminants in different cereals and edible products; therefore, different studies have been focused on the implementation of methodologies for detection, analysis, and quantification of these compounds. In this context, there is still a vast diversity of alkaloids derived from psychoactive mushrooms that constitute an unimaginable source of bioactive molecules, as well as inspirational templates for the search and design of new drugs.

### 4.3. Analytical Methods and Alternative Strategies of Production

Because of the vast complexity and high variability in the chemical composition of psychedelic fungi, there is no single method or standardized protocol for the detection and quantification their derived metabolites. Instead, several analytical methods have been reported for the analysis of psychedelic alkaloids in mushrooms samples including gas (GC) and liquid (HPLC) chromatography coupled with mass spectrometry (MS), diode-array detection (DAD), or fluorescence (FL). This information constitutes the basis for the development of new procedures for a specific sample in the current and future research of psychoactive alkaloids. Considering that information related to the analysis of psychoactive alkaloids have been comprehensive addressed and discussed in previous publications [18,65], in this review, some of the most representative and innovative examples have been tabbed and summarized in Table 4.

The preeminent technique for the qualitative and quantitative analysis of indole alkaloids such as tryptamines, ergolines, and β-carbolines is reverse phase-based liquid chromatography (LC), attributed to their predominantly polar nature [20]. Most of the LC methods reported for the analysis of mushroom-derived indoles use C_18_ columns with polar mobile phases (MP) composed by an aqueous phase (generally buffered) and organic solvents such as methanol or acetonitrile (Table 3). The mobile phases can be used either in isocratic or gradient mode, and the selection of the accurate program depends on the matrix complexity and should be determined experimentally [20,67]. In terms of detection methods, one of the most used was Diode-Array Detection (DAD), since the aromaticity and scaffold diversity of indole alkaloids allow for their monitoring at variable wavelengths. For instance, tryptamines are usually detected around their λ max 280 nm, whereas carbolines have a variable λ max absorption between 300 nm and 340 nm [50,74]. Moreover, thanks to the notable advances in mass spectrometry and its coupling interfaces with chromatographic techniques, LC-MS has become an essential tool in natural products chemistry, allowing the simultaneous analysis of metabolites with diverse polarities in lower detection limits [75]. In the case of psychedelic indole alkaloids, the use of electrospray ionization (ESI) in positive mode with low- or high-resolution analyzers such us quadrupoles (Q), orbitrap, triple-quadrupoles, and quadrupole-time of flight (q-TOF) has been mainly reported (Table 4). In addition, the use of cutting-edge techniques such as high-resolution matrix-assisted laser desorption/ionization mass spectrometry (HR- MALDI-MS imaging) has been implemented to screen and track the production of carbolines both in fungal mycelium cultures and fruiting bodies [50,71]. MALDI imaging is a valuable tool for the spatial detection of chemical compounds in complete samples without prior extraction resulting in fast and efficient screening that allows for target isolations. For instance, this technique was successfully used for the detection of β-carboline alkaloids in the fruiting bodies of *Mycena metata*, leading to the isolation of different metatacarbolines, one of those with a new chemical structure [71].

#### Alternative Strategies of Production

Since traditional methods of extraction and synthesis of most of the psychedelic alkaloids are laborious, expensive, and unattainable for drug development, the interest in alternative sources of production of these alkaloids has become evident to supply the amount of active ingredient for required trials, pharmacological studies, and the possible large-scale production with a view to commercialization. Although mushroom-derived natural products have the advantage of being obtained from easily cultivable fungal material, the extraction and purification process are usually time-consuming and might yield few milligrams of the bioactive compounds. In addition, the production of active alkaloids from fungal biomass implies other challenges such as high variability in yields because of differences between batches and low stability of the molecules, hindering the scalation of extraction process at the industrial level [76]. On the other hand, synthesis, which is the most widely used alternative at large-scale production of bioactive molecules, is not the most suitable replacement to produce psychoactive-alkaloids, since chemical production is usually arduous, involving multistep reactions or expensive regents. For example, the synthetic routes for simple tryptamines, such as psilocybin involve four-steps reactions, and in some cases requires the use of toxic or expensive reagents. Additionally, the yield of the total synthesis of psilocybin currently reported in the literature is low, implying a high lab-production cost which is estimated around USD 2000 per gram of the alkaloid [77]. Furthermore, in the case of complex molecules such as ergot alkaloids containing tetracyclic scaffolds and several stereo-centers, the synthetic landscape is more challenging. In fact, the asymmetric synthesis of lysergic acid was accomplished only one decade ago, and asymmetric routes with scalable potential were described only in the last few years [78,79]. Therefore, in view of the vast therapeutic potential of mushroom-derived psychedelic alkaloids, more suitable and efficient alternatives for larger-scale production have been explored, and among them biotechnological production, biocatalytic, and hybrid lab-synthesis are the most recent advanced approaches.

Biotechnological production of bioactive psychedelic alkaloids has been driven by advances in omics techniques facilitating the elucidation of biosynthetic pathways, gene clusters, and enzymes involved in their bioengineering [77,80]. For example, enzymes such as decarboxylase (PsiD), kinase (PsiK), and methyltransferase (PsiM) have been employed to transform in vitro 4-hydroxy-l-tryptophan into psilocybin [77]. Recently, hybrid synthetic-biocatalytic routes have been explored as a scale-up alternative, with one or more critical steps in the chemical-synthetic route being replaced by biocatalytic reactions using recombinant enzymes. Indeed, in psilocybin synthesis, the phosphorylation of the substrate is the limiting step, requiring multiple reactions, expensive reagents, and having poor atom economy. However, it has been reported that when using a recombinant mushroom kinase, this phosphorylation can be accomplished within 20 min yielding milligrams of the active alkaloid, turning this approach into an interesting upscaling option [81]. On the other hand, in vivo biosynthesis has also been studied to produce psilocybin, by genetic engineering of a microbial host such as *Aspergillus nidulans*, *Escherichia coli*, and *Saccharomyces cerevisiae*. In this area, one of the most recent achievements was the production of psilocybin using recombinant *S. cerevisiae* engineered with *P. cubensis* genome and booting the pathway with a novel cytochrome P450 reductase. The fed-batch fermentation of this modified *S. cerevisiae* yielded the production of 627 mg/L psilocybin and 580 mg/L of psilocin, along with traces of other tryptamine derivatives such as baeocystin and norpsilocin, and proving an appropriate engineering of the biosynthetic pathway [82]. Nonetheless, the yields and costs of biosynthetic production of mushroom-derived alkaloids with therapeutic relevance continue to be a challenge for scaling up the production, and more studies are required to optimize the production conditions of these metabolites on a large-scale.

## 5. Psychopharmacological Potential of Mushroom-Derived Indoles

Historically, natural alkaloids have made a significant contribution to drug discovery by providing leads, candidates, and inspiring scaffolds for the design of new drugs. Moreover, numerous bioactive alkaloids exert distinctive multi-pharmacological properties which make them suitable candidates in the search for psychotherapeutic agents. Indeed, psilocybin, DMT, and LSD are just a few examples of alkaloids that have prompted the reviving of psychedelic research programs in recent years [10,16,18,40]. Even though the pharmacology of mushroom-derived alkaloids has been under-explored because of governmental and legal restrictions, preliminary preclinical studies have provided encouraging results, and have revealed interesting pharmacological traits of those natural compounds. These findings might constitute a crucial starting point to promote future therapeutic and clinical research with natural psychedelics, since strong evidence of their translational medicine potential is still missing [20,21]. Herein, the neuropharmacological potential of different natural isoquinolines are summarized and discussed, encompassing in vitro activity, molecular targets, structural motifs, and preclinical studies (Table 5).

### 5.1. Preclinical Research

The outstanding psychopharmacological activity of indole alkaloids lies in their structural similarity with different neurotransmitters, specific structural motifs shared with the biogenic amines that allow them to interact with receptors and transporters in the nervous system triggering different biological responses [83]. For instance, tryptamine derivatives share the same scaffold of serotonin (5-hydroxytryptamine), a monoamine neurotransmitter that plays complex physiological role through the regulation of multiple functions such as perception, cognition, memory, emotion, and appetite, among others. Consequently, tryptamines are essentially recognized for their ability to modulate serotonin receptors, endowing them with unique psychotropic and pharmacological properties [25]. Psilocybin is the major tryptamine found in magic mushrooms, commonly recognized for being the active alkaloid involved in the hallucinogenic effects by activation of the serotonin 2A receptor (5-HT_2A_). A vast number of preclinical and clinical studies have uncovered the promising pharmacological potential of psilocybin in the treatment of neurological conditions, including anxiety, depressive disorders, chronic pain, and neuroinflammation, among others [19,38,82,84]. However, it has been demonstrated that psilocybin acts as a pro-drug, being rapidly dephosphorylated by alkaline phosphatases in the stomach, intestines, and blood, forming psilocin, which is responsible for the psychedelic and therapeutic properties [82]. Although the exact molecular mechanisms underlying the neuropharmacological activity of psilocybin are not completely understood, preliminary pharmacodynamic findings indicate that psilocin targets primarily serotoninergic receptors, simultaneously triggering the dopaminergic and glutaminergic responses by modulating other therapeutic targets [85]. In vitro studies using 5-HT humans or mice receptors have also demonstrated that psilocybin has lower agonist potency in comparison to psilocin (Table 5). In fact, psilocin showed agonist activity at 5H-T_2A_ receptors levels with effective concentrations in the nanomolar range, whereas its phosphorylated analog was about thousand times less potent [68,86]. The 5H-T_2A_ receptors are predominantly expressed in the visual and orbitofrontal cortexes; consequently, their stimulation is associated with the changes in visual perception and colorful hallucinations characteristic of psychoactive substances [85]. Moreover, it has been suggested that the neuropharmacological activity of psilocybin involves a multi-target action through the modulation of several receptors in the central nervous system (CNS) with variable affinities, which have been supported with in vitro studies. Psilocin modulates multiple serotonin receptors (5H-T_1D_, 5H-T_1E_, 5H-T_2C_, 5H-T_2B_, 5H-T_5–7_), as well as at adrenergic (α2A and α2B) and dopamine receptors (D_3_) with *Ki* values between the nanomolar to millimolar range (Table 5).

In vitro studies have not provided a conclusive understanding of psilocybin pharmacology, hinting that its mechanism of action might be highly sophisticated and exceptional. For this reason, multiple in vivo models have been employed for a better perspective of the pathways, interconnections, and targets involved in the therapeutic effects of this tryptamine. For example, Heseelgrave et al. [100], showed that a single injection of psilocybin (e.g., 1 mg/kg) in chronically stressed mice can reduce the anhedonic behavior in the forced swim and sucrose preference assays. This study also indicates that the antidepressant-like effects of psilocybin in this model are strongly linked to the restoration of the synaptic activity in the cortico-mesolimbic region, which is important for the assimilation of rewards and emotions. Additionally, their findings suggest that the anti-anhedonic effects of psilocybin could be independent of the 2A serotonin receptors activation since the response in the behavioral and electrophysiological test was not affected by the 5-HT_2A/2C_ antagonist ketanserin [100]. Similarly, a single intraperitoneal injection of psilocybin (1 or 2 mg/Kg) reduced the compulsive-like behavior in mice after 15 min of administration in the marble burying assay. The pre-treatment of the animals with a 5-HT_2A_ antagonist (M100907) did not block the psilocybin effects in the digging behavior, which suggests the anti-compulsive activity of this alkaloid is not exclusively related to the modulation of 5-HT_2A_ receptors [101]. Other in vivo studies showed that activation of 5HT_2A_R produces changes in the psychosis-like head-twitch response (HTR), and this behavioral assay is widely used to estimate the 5HT_2A_ antagonist activity and hallucinogenic effects in animal models. In this test, the side-to-side movement of mice heads was increased after the administration of different psilocybin doses. In addition, the pre-treatment with selective 5-HT_2A_ antagonist suppressed the effects of the alkaloid in the head-twitch response test, supporting the in vitro findings on the agonist activity of psilocin with 5HT_2A_R [102].

Although the pharmacological role of 5-HT_2A_R is still not fully elucidated, strong scientific evidence has ratified its implication in the neuropsychological effects of psychedelics in both animals and humans [83,89]. On the other hand, the abnormal function of 5-HT_2A_R have been associated with different neuropsychiatric disorders such as anxiety, compulsive-disorders, depression, and schizophrenia, among others [103,104]. Thus, from the pharmacological perspective, the ligands targeting the regulation of the activity or expression of this receptors represents a valuable therapeutic option for neuropsychiatric and neurodegenerative disorders. Pharmacological evidence suggests that the therapeutic potential of psilocybin on neuropsychiatric disorders is associated with a multimodal action and downstream signaling effects, and not only by modulation of serotonin receptors. For example, Shao et al. [105] determined that a single dose of psilocybin (1 mg/kg) mitigated the stress-induced behavior and increased the excitatory neurotransmission in mice. Interestingly, this study demonstrated that psilocybin therapy produces favorable alterations in the synaptic network by increasing the formation rate of dendritic spines and the spine size in the frontal cortical pyramidal cells in mouse brains. The synaptic rewiring in the cortex triggered by psilocybin was observable within 24 h and had a long-lasting effect. Furthermore, this restoration went along with enhanced excitatory neurotransmission. Similar effects were reported in a recent study, in which changes in the synaptogenesis of the pig brain after treatment with psilocybin were studied [106]. After being injected with psilocybin (0.08 mg/kg), changes in the hippocampus and prefrontal cortex of the animals were analyzed using autoradiography at 1 or 7 days after treatment. The results of this study showed an intensification in the synaptic vesicle protein 2A and a considerable reduction in the 5-HT_2A_R density in the hippocampus at one day post-treatment. However, after seven days, a regulation in the density of 5-HT_2A_R was observed, whereas the synaptic vesicle was still significantly higher. The findings of the above in vivo studies ratify the crucial role of synaptogenesis induced by psilocybin, and its potential close relationship with the therapeutic effects in the management of neuropsychiatric diseases, through the promotion and reconnection of neuronal networks in different regions of the brain. In addition, other downstream effects proposed for psilocybin include the regulation of important neurotransmitter receptors (such as GABA, AMPA and NMDA), as well as the activation of factors involved in neuronal plasticity such as brain-derived neurotrophic factor (BDNF), tumor and necrosis factor-alpha (TNF-α) [85]. Therefore, taking all those effects together, it is possible to conclude that psilocybin-therapy involves a unique and revolutionary mechanism of action with long-term effects, which would explain some of the preliminary results obtained in patients in ongoing clinical trials.

The remarkable neuropharmacological potential of psilocybin (psilocin) in in vitro and in vivo studies with other mushroom-derived tryptamines are limited, which is mainly attributed to the lack of availability of pure compounds obtained from fungal material. Hence, chemical synthesis has been explored to supply larger amounts of these alkaloids in order to undertake further pharmacological research, and structure-activityrelationship (SAR) studies to identify possible key motifs that can be useful in designing new bioactive molecules [68,86,107]. Recently, Sherwood et al. [68] reported the synthesis of norbaeocystin, baeocystin, norpsilocin, and aeruginascin using a multi-step route with yields between 60% and 90%, as well as the subsequent in vivo and in vitro assays with baeocystin to assess its putative hallucinogenic effects and modulatory activity in 5-HT_2A_ receptors. In an in vivo test, mice were injected with variable doses between 0.03 to 3 mg/kg of baeocystin and the head-twitch response (HTR) measured for 20 min, without detecting significative psychedelic-like effects. Then, the in vitro activity on the 5-HT_2A_ receptors was assessed using norpsilocin, the active metabolite that would be generated from the in vivo dephosphorylation of baeocystin. Norpsilocin showed full agonist activity at human and mice 5-HT_2A_ receptors with a similar potency of psilocin (Table 5), corroborating that metabolic or pharmacokinetic factors might be involved in the absence of in vivo activity of baeocystin. Similar findings in the HTR assay were reported with norbaeocystin, a N-demethylated analog of baeocystin and psilocybin. After oral administration of this tryptamine, no significant effects in the head-twitch response were observed, indicating a low capacity of this alkaloid to target the 5-HT_2A_R in the CNS [108]. The above results demonstrate that despite the high structural similarity between psilocybin, baeocystin, and norbaeocystin (Figure 4), their pharmacological profile differs ostensibly, suggesting that N-methylation degree in the lateral chain of tryptamines might play a critical role in their in vivo pharmacokinetic or pharmacodynamic properties. Indeed, it has been suggested that the fast metabolic deamination by monoamine oxidases (MAO) of baeocystin, norbaeocystin, and other related tryptamines would be related to the limited in vivo activity. Since primary and secondary amines are more likely metabolized by MAO, baeocystin and norbaeocystin might undergo first-pass metabolism before reaching the site of action in the CNS [68].

In the case of bufotenine, a tryptamine historically recognized for conferring hallucinogenic effects to different species of toads, Amazonian plants, and *Amanita* mushrooms, there is general controversy around its psychoactive properties. The literature reports regarding the pharmacological activity of bufotenin are ambiguous, inconclusive, and occasionally contradictory. In vitro studies with this tryptamine have demonstrated potent agonist activity at different serotonin receptors including 5-HT_2A_, 5-HT_2C,_ 5-HT_1A_, 5-HT_1B_, and 5-HT_3_ (Table 5). The in vitro results suggest that bufotenine has comparable or higher potency than psilocin with some of those serotonin receptors [88]. Bufotenine is a N-methylated analog of serotonin and positional isomer of psilocin, so it appears logical to assume these structural similarities are linked to its potent agonist activity with 5-HTRs. Additionally, in silico molecular docking has shown that bufotenine and LSD might have identically affinity with 5-HT_2A_, establishing strong interactions (i.e., H-bonds and ionic-bond) with the same aminoacidic residues in the binding site [88]. Nonetheless, the in vivo pharmacological profile of bufotenine is markedly different for psilocybin; for example, it has been reported that a 10-fold dose higher of bufotenine is required to induce behavioral alterations similar to psilocybin in rats [88]. Moreover, although 4 mg of a single oral dose of psilocybin is sufficient to induce noticeable psychotropic effects, 100mg of bufotenine does not generate perceptible mental alterations in humans [109]. The lack of in vivo activity of bufotenine has also been attributed to its physicochemical properties, mainly to its poor ability to cross the blood–brain barrier (BBB) caused by low lipophilicity. Moreover, other studies have also shown bufotenine is quickly metabolized by MAO-A after its injection in rats, which would also lead to a low concentration of the alkaloid and hindering BBB penetration [110]. Because of the close structural similarity between psilocin and bufotenine (Figure 4), it seems unlikely to attribute the absence of in vivo activity of bufotenin to its pharmacokinetic features, since those tryptamines differ only in the position of the hydroxyl group. Recently, Lenz et al. [109], through a series of nuclear magnetic resonance (NMR) experiments and quantum chemical calculations, have displayed the fundamental role played by the hydroxy group on the C4 position of the indole scaffold in the distinctive pharmacology traits of psilocybin. The NMR spectroscopy analysis showed that psilocin can establish an intramolecular hydrogen bond between the hydroxyl group of C4 and the N of the aminoethyl sidechain, leading a pseudo-ring formation that in turn provides unique physicochemical features to this alkaloid (Figure 4). Indeed, the pseudo-ring formation decreases the amino group ionization, increasing the lipophilicity and the partition coefficient, and conferring a high ability to permeate the BBB [109]. Contrarily, 5-hydroxytryptamines such as bufotenine cannot establish the intramolecular hydrogen bond and this would be the reason for their weak pharmacokinetic properties. The above findings are not only an example of the power nature in the design of bioactive molecules with privileged traits, but also these structural clues are also a big step for the rational modeling of synthetic analogs for the development of new psychotherapeutic agents.

Furthermore, β-carbolines such as harmine, harman, tetrahydroharmine (THH), and harmaline recently detected in mushrooms of the *Psilocybe* genus have proven to have an interesting profile of pharmacological activity related to neuropsychiatric disorders [70,90,111,112]. Harmine, harmaline, and THH are also popularly known as ayahuasca alkaloids owing to their widespread presence in hallucinogenic beverages (i.e., ayahuasca tea) prepared by indigenous communities of northwestern South America for magical-ritual practices. Natural β-carbolines have demonstrated multi-target pharmacological activity by modulation of enzymes and receptors involved in the pathophysiology of depressive disorders such as monoamine oxidases (MAO), serotonin, and dopamine receptors. β-carbolines such as harmane, harmine, and harmaline present potent inhibitory activity against MAO-A and no significant inhibition against the B isoform of this enzyme. For example, harmine and harmaline showed reversible inhibition of MAO-A with an IC_50_s lower than 15 nM, whereas their analog, harman, possesses lower inhibitory activity with an IC_50_ higher than 300 nM [90]. MAO enzymes play an essential role in brain function by regulating concentrations of biogenic monoamines such as serotonin, dopamine (DA), and norepinephrine. An imbalance in MAOs activity entails drastic changes in neurotransmitter levels, which have been identified as a common hallmark in the pathophysiology of different psychiatric disorders, including chronic stress, major depressive disorder, and alcohol dependence. Indeed, in depressive disorders, the MAO-A hypothesis links the upregulation of this enzyme with lower levels of 5-HT, DA, and noradrenaline, neuronal damage, and reduction of cortical volume [113]. Hence, selective MAO-A inhibitors such as β-carboline alkaloids are promising candidates for the discovery of new antidepressant drugs. Moreover, it has been theorized that β-carbolines might play a potential synergistic role with other psychoactive tryptamines in mushrooms extracts and shamanic beverages, considering that MAO-A is the enzyme responsible for the first-pass metabolism of bioactive tryptamines before reaching the CNS. From this perspective, it would be critical to undertake pharmacological studies to evaluate the possible differences in the metabolism and bioavailability of active tryptamines such as baeocystin, norbaeocystin, and psilocybin in combination with β-carbolines. Likewise, it would be interesting to evaluate the changes in the neuropharmacological activity of *Psilocybe* mushrooms with variable content of β-carboline alkaloids.

Even though the underlaying mechanisms involved in the antidepressant activity of β-carbolines are not fully elucidated, multiple preclinical studies with harmane and harmine have reinforced the neuropharmacological potential. In in vivo models, these alkaloids have shown regulatory activity on serotoninergic, dopaminergic, and GABAergic signaling pathways, as well as downstream activity in neurogenesis, neuroplasticity, astrocytic restoration, and synaptogenesis by modulation of important brain proteins. For example, harmane has been reported to exert antidepressant and anxiolytic effects at different doses in rats and mice. In those models, the acute intraperitoneal administration of harmane induced a reduction in the depressive-like behavior in a dose-dependent manner [92,94]. Such behavioral changes have been correlated with the downregulation of the monoaminergic system, leading to an increase in serotonin levels in the hippocampus, hypothalamus, amygdaloidal cortex, and prefrontal cortex [90,92,111]. Similarly, harmine has shown antidepressant potential in animal models using chronic unpredictable stress (CUS) and chronic mild stress (CMS) protocols. In both cases, after intraperitoneal administration of harmine, (for 7 and 10 days, respectively) a significative reduction in the depressive-like behavior was shown [95,96]. In the CUS protocol, C57BL/6J mice treated with 20 mg/kg of harmine showed lower levels of brain-derived neurotrophic factor (BDNF) protein and increased expression of glutamate transporter 1 protein (GLT-1) compared to the control; these results suggest that harmine acts as antidepressant by restoration of astrocytic functions in mice [95]. Moreover, the treatment of male adult Wistar rats with harmine (15 mg/kg × 7 days) after the CMS protocol reversed the anhedonic behavior and recovered the damage in the adrenal gland caused by chronic stress [96]. These initial preclinical findings revealed that some of the β-carbolines presented in magic mushrooms can exert antidepressant and anxiolytic effects through the modulation of multiple CNS targets. Nevertheless, it is important to emphasize that further molecular and toxicological studies are still required to validate the real “drug-likeness” potential of these indole alkaloids. Perhaps one of the main drawbacks of these alkaloids at the clinical level might be their low oral bioavailability owing to first-pass metabolism since previous pharmacokinetic studies have proven that β-carbolines are rapidly oxidated in vivo by different P450 enzymes, decreasing their half-life and distribution profile [90]. At the toxicological level, the main disadvantage in the clinical applicability of these alkaloids lies in the possible off-target effects associated with their intercalation with DNA, which could involve serious safety issues because of the chronic administration of β-carbolines. Structural features such as the fully tricyclic aromatic scaffold and coplanar methoxy substituents have been correlated with their capacity to establish strong interactions with nucleic acids, potentially linking these characteristics to neuropharmacological activity [70]. Therefore, preclinical safety and toxicology studies are imperative to determine the drawbacks after long-term administration of carbolines.

Finally, natural ergot alkaloids (EA) and their derivatives have demonstrated interesting poly-pharmacological properties in preclinical research with several CNS-related diseases such as dementia, Parkinson’s disease, chronic pain, and migraines. Structural activity studies have postulated that the therapeutic spectrum of ergots resides in their pharmacophoric relationship with various monoamine neurotransmitters, such as dopamine, serotonin, and adrenaline, among others [98]. However, the accurate mechanisms behind the medicinal traits of ergot alkaloids are not completely elucidated yet, mainly because of the great complexity of the central and peripheral nervous systems, and the gap in the pathophysiology of neurological disorders. Even though mushroom-derived ergot alkaloids present interesting pharmacological characteristics, the semi-synthetic derivatives have had a higher impact on modern pharmacology since they have been approved for the treatment of different diseases. The most representative examples of natural EAs with pharmacological relevance are ergotamine and lysergic acid, both characterized by their agonist activity in serotonin, dopamine, and noradrenaline receptors. As exemplified in Table 5, ergotamine has a potent affinity for different serotonin receptors, including 5-HT_2A_, 5-HT_2B_, 5-HT_1A_, and 5-HT_1B_, with comparable or even higher potency than psilocybin. However, different in vivo models showed that high oral doses of the ergotamine are needed to generate therapeutic effects, which resulted in the main drawback of this alkaloid at the pharmacological level. Indeed, further studies demonstrated that the low oral bioavailability of ergotamine (<1%) is caused by a high first-pass metabolism. In addition, ergotamine has also shown different side-effects related to its multi-target properties, ranging from acute undesirable effects (i.e., muscle pain, nausea, numbness, and tingling of the fingers and toes) to serious cardiovascular problems (i.e., coronary vasoconstriction, ischemic changes, and anginal pain) [99]. Thus, despite the exuberant in vitro activity of some mushrooms-derived EAs, their clinical application has been overshadowed by their off-target effects, nevertheless, the greatest contribution of the natural EAs in pharmacology has been to provide templates for the design of new synthetic derivatives. Some examples include dihydroergotamine, a saturated derivative of ergotamine, that presents higher potency against adrenergic receptors and better pharmacokinetic characteristics and is prescribed for the treatment of acute migraine attacks. α-Dihydroergocryptine, a derivative of ergocryptine, is a dopamine agonist used as monotherapy in the early stages of Parkinson’s disease. Nicergoline, a synthetic derivative of D-lysergic acid, is used in the treatment of dementia and vascular disorders such as cerebral thrombosis. Perhaps the most representative example of ergot derivatives is the D-lysergic acid diethylamide or LSD, a synthetic analog of lysergic acid widely recognized by its psychoactive properties linked to its potent agonist activity for the 5-HT_2A_ receptor (K*_d_*  = 0.33 nM). Moreover, LSD is nearly 100 times more potent than psilocybin against most serotonin, adrenaline, and dopamine receptors [114]. Likewise, psilocybin has shown remarkable neuropharmacological activity in animal models. However, it should be noted that similar to other ergot alkaloids, the promiscuous polypharmacological activity of LSD might lead to off-target effects in prolonged uses, compromising its safety profile. On the other hand, one of the main therapeutic advantages of psilocybin compared to LSD lies in the higher oral bioavailability and shorter intervals of the psychedelic effects, which would enable the management of patients at the clinical level in shorter intervention sessions.

### 5.2. Ongoing Clinical Trials

Psilocybin is the mushroom-derived indole alkaloid with the highest contribution in clinical trials for neuropsychiatric disorders. In fact, more than 100 psilocybin-related clinical trials are currently registered on the ClinicalTrials.gov data base (https://clinicaltrials.gov accessed on 10 October 2022) and another 16 on the European Union Clinical Trials (https://www.clinicaltrialsregister.eu/ accessed on 10 October 2022)). Regarding the progress of clinical studies registered on ClinicalTrials.gov, 26% are in recruitment phase, 36% have not started the recruitment stage yet, 14% are in course, and just 19% have been completed. These figures indicate that there is still some way to go before the conclusive clinical outcome of psilocybin; nevertheless, the clinical landscape is also optimistic considering the growing research engagement in the therapeutical potential of this tryptamine. As shown in the Figure 5, about 55% of those ongoing clinical trials comprise mental health conditions, including neurological, neurodegenerative, and neuropsychiatric disorders (Figure 5A). Within the mental disorders, most of the ongoing clinical research involves psilocybin-assisted therapy for the management of depressive, substance-dependence, and anxiety conditions, accounting for more than 60 studies (Figure 5B).

Selected examples of the ongoing psilocybin-related clinical trials for depression, anxiety, and substance-dependence disorders are listed in Table 6. The current clinical research lies predominantly in psilocybin-assisted psychotherapy with a conventional experimental design that involves dosing sessions or drug intervention, and usually one or two sessions assisted by healthcare professionals, as well as complementary sessions of psychological support therapy. It should also be noted that of the total clinical studies reported, only around 20 have been completed, and among them the majority correspond to preliminary phases (0 and 1), where pharmacokinetics, dosage, and/or safety were evaluated in healthy volunteers. In escalating doses studies, psilocyn and psilocin showed a linear pharmacokinetic profile after oral administration of 0.3, 0.45, and 0.6 mg/kg in healthy adults. Psilocybin was not detected in plasma or urine because of its fast first-pass metabolism, and the elimination half-life of psilocin was 3 h with lower renal clearance (<2%). In addition, psilocybin has demonstrated a favorable safety profile without toxicity reports and no significant adverse effects even at the highest dose tested (0.6 mg/kg) [115]. Moreover, in the follow-up studies after therapeutic intervention with psilocybin, 94% of the healthy participants reported beneficial effects described as a life-changing experience, as well as improvements in mood, mindset, gratification, and social relationships [82,116].

The clinical research of psilocybin relating to depressive disorders have provided encouraging preliminary outcomes. For example, patients with major depressive disorder (MDD) treated with two doses of psilocybin (20 or 30 mg/kg) separated by 1.6 weeks showed a significative increased cognitive and neural flexibility, as well as reduced concentrations of glutamate and N-acetylaspartate in the anterior cingulate cortex in the brain [38]. In another study, 24 patients with moderate to severe MDD were treated with two doses of psilocybin with supportive psychotherapy followed through 12 months. All of them showed a considerable decrease of the MDD baseline behavioral symptomatology, and experienced other well-being effects. Furthermore, no participants reported voluntary consumption of psilocybin outside of the study and no serious adverse effects were observed during the clinical trial [39]. Common findings for the concluded clinical trials for depressive and anxiety disorders strongly suggest that psilocybin-therapy is more effective when combined with psychotherapy rather than just drug intervention. Thus, based on those important outcomes, most of the clinical studies in the recruitment phase (or have not started yet) have included in their experimental design two main stages combining the therapeutic intervention and assisted psychotherapy. Moreover, psilocybin has shown the most promising clinical outcomes in the treatment of depressed patients, which has shown sustained remission at treatment with several antidepressant drugs, some of them even with suicidal behaviors. For this reason, psilocybin was designated by the U.S Food and Drug Administration (FDA) as a “breakthrough therapy” for treatment-resistant depression in 2018, given a historical step forward in the use of psychedelic alkaloids for neuropsychiatric diseases.

Furthermore, it is worth pointing out that a considerable percentage of the clinical trials involving anxiety and depression are concentrated in the management of mood symptoms triggered by chronic life-threatening diseases such as cancer and Parkinson’s (Table 6). In fact, Griffiths et al. [117] demonstrated the efficacy of psilocybin in the reduction of depression and anxiety symptoms in terminal cancer patients. High doses of psilocybin (22 or 30 mg/70 kg) drastically reduced anxiety and depression parameters in treated patients, and 80% of the participants showed long-lasting beneficial effects at least until the 6-month follow-up session. Cancer patients reported relief in the anxiety associated with death, detachment from disease, spiritual experiences, and emotional reconnection with life [117]. Psilocybin-therapy, unlike pharmacotherapies for managing symptoms in cancer patients (e.g., opioids), presented lower side-effects, and sustained well-being outcomes, which in turn results in a better quality of life. Therefore, psilocybin and other psychedelic alkaloids constitute a valuable therapeutic alternative in end-of-life and palliative care, an area hardly explored at the pharmacological level and a major concern for global public health considering the growing number of cases of patients with complex chronic diseases.

An additional plausible therapeutic use of natural psychedelic indoles is the treatment of substance dependence disorders. In fact, this is the second largest category of the reported ongoing clinical trials for psilocybin (Figure 5) with around 20 studies, ranging from the treatment of nicotine and alcohol dependence to the abuse of amphetamines and opioids. For example, exploratory studies for the treatment of alcohol dependence have shown a significant reduction in the behavioural consumption in the participants psilocybin-assisted psychotherapy compared to the control group [118]. Even though these results appear promising, there is still a lack of solid evidence in the translational use of psilocybin and other psychedelics for the management of substance dependence disorders. Considering this application involves patients with a high risk of substance abuse, the completion of the clinical trials in progress would be crucial to gather robust data to support this therapeutic application.

#### Safety Profile

Undoubtedly, the main safety concern in the therapeutic use of mushroom-derived alkaloids is the possible addictive liability linked to their psychotropic nature. A long-term safety profile and dispensing management are crucial points for the approval and commercialization of natural psychedelics for therapeutic purposes since patients with neuropsychiatric diseases are more susceptible to present drawbacks such as paranoia, suicidal ideation, and dependence [82]. In this context, the future clinical research should be focus on the evaluation of the long-term addictive potential, adherence, and behaviour side effects. Thus far, most of the clinical studies with psychedelics have shown a lower risk of addictiveness and minor safety issues [34]. However, it is worth mentioning that most of these studies have been carried out in controlled conditions, with small groups, and in short-term periods of time. Thus, robust clinical evidence should be gathered to ensure a broad safety spectrum, which will only be possible through the strengthened of R&D programs with higher government, legislative, and funding support.

For psilocybin, the available results of clinical trials have indicated a favorable safety profile with minimal side effects, low toxicity, and no reports of addictive engagement, even up one year after the first treatment. Moreover, it has been reported that multiple doses of psilocybin (four at least) generated tolerability but did not trigger withdrawal symptoms or physical dependence. Ultimately, the concerns related to the consumption and adequate management of natural psychedelics at a therapeutic level can be addressed through assisted-therapy programs, ensuring a safety medical and professional setting before, during, and after treatment.

## 6. Conclusions

Based on ancestral knowledge and solid scientific evidence, it is possible to assert that natural and synthetic psychedelics are nowadays one of the most powerful and potential tools for the discovery of new therapeutic agents to treat complex neuropsychiatric disorders. Therefore, it is still necessary invest in research programs to search for new sources of natural hallucinogens, and also to design innovative administration and release routes, as well as studies to understand the chemistry, pharmacology, and toxicology of promising psychedelics. Despite the outstanding progress in analytical methods of secondary metabolites, the current isolation and chemical identification approaches still represent a barrier in the advancement of phytochemical screening of mushroom-derived psychedelic alkaloids. Therefore, scientific, legislative, and funding efforts are still required to promote innovative chemical and pharmacological studies employing novel and forefront methods to reduce the bottlenecks in psychedelic natural products-based drug discovery. For example, strategies involving the use of metabolomics in conjunction with molecular networking and bio-chemometrics may improve the detection and targeted-isolation of bioactive metabolites.

In the past decade, encouraging preliminary outcomes on the use of psilocybin and other mushroom-derived indole alkaloids to effectively treat depressive disorders have fueled the interest in psilocybin-assisted psychotherapy. Clinical research has focused on the application of psilocybin to treat depressive disorders, such as anxiety and depression, particularly in the context of mood symptoms triggered by chronic and terminal diseases. Overall, psilocybin and other psychedelic alkaloids have shown the most promising clinical results in the treatment of several mental disorders, which are a substantial long-term challenge for the global health system. Because of their rising prevalence, uncertain neuropathology, and lack of effective pharmacological treatments, mental illness is a major concern for global public health, and indole alkaloids, particularly those derived from magic mushrooms, highlight the promising application of such active ingredients as safe and effective therapeutic agents for the treatment of neuropsychiatric disorders.

## Figures and Tables

**Figure 1 biomedicines-11-00461-f001:**
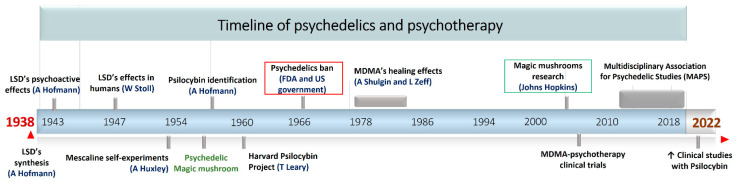
Timeline highlighting main historical events relating to the use of psychedelics in psychotherapy.

**Figure 2 biomedicines-11-00461-f002:**
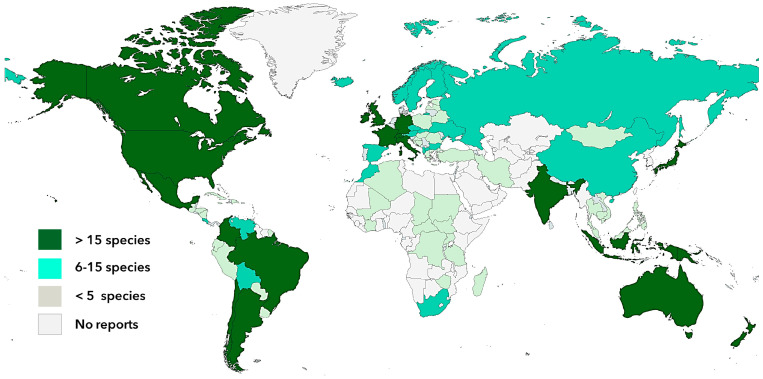
World geographic distribution and diversity of psychoactive mushrooms.

**Figure 3 biomedicines-11-00461-f003:**
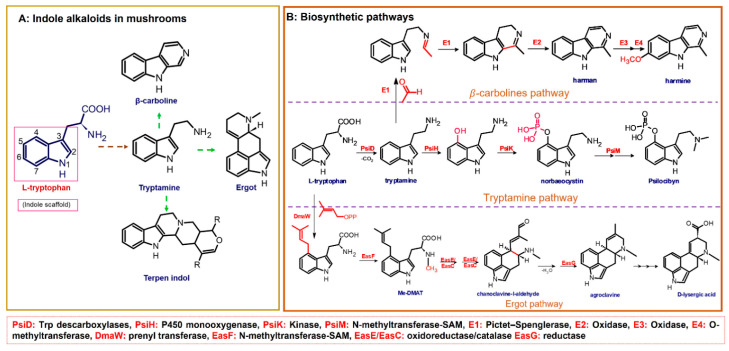
(**A**): Subclasses of indole alkaloids reported in mushrooms, (**B**): general overview of some fungal biosynthetic pathways for indole-derived alkaloids.

**Figure 4 biomedicines-11-00461-f004:**
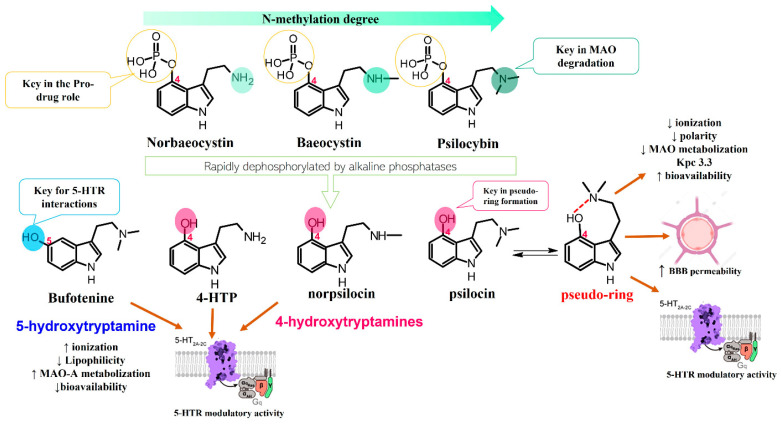
Important structural motifs of tryptamines and their pharmacodynamic and pharmacokinetic correlation in neuropharmacology.

**Figure 5 biomedicines-11-00461-f005:**
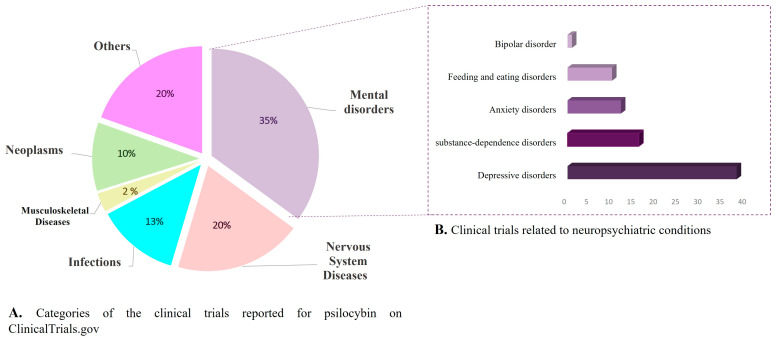
Distribution of psilocybin-related clinical trials registered on the ClinicalTrials.gov data base (https://clinicaltrials.gov accessed on 10 October 2022).

**Table 1 biomedicines-11-00461-t001:** Selected examples of psychoactive mushrooms with ethnopharmacology relevance around the world.

Scientific Name	Common Name	Community (Country)	Uses	Ref.
*Amanita muscaria*	*Miskwedo*	Ojibwa and Algonquin (North America)	Used in sacred ancestral ceremonies and for the treatment of mental and spiritual ailments, supplying states of happiness and relaxation.	[27,28]
*Cordyceps sinensis*	*Chong xia cao*	Tibet (China)	In traditional Chinese and Tibetan medicine, this fungus was recognized for its aphrodisiacal properties. In addition, alcoholic extracts of this mushroom are used for mood improvement and as energy drinks because of its ginseng-like effects.	[29]
*Gymnopilus* *purpuratus*	*Badiadimurobuni* *The ear of the spirit*	Amazonian natives (Brazil)	In the XVII century, Jesuits reported that tribes of central Amazonia in Brazil, Peru, and Ecuador prepared *Gymnopilus*-based inebriating beverages for shamanic and healing purposes.	[21]
*Inocybe* *aeruginascens*	*Fibrous head*	No specified (Hungary)	This mushroom is also recognized in different Europe countries as the “good trip”, because after its consumption people describe an exceptionally pleasant sensation with sparkling fantasies, experience of a flying soul and euphoric feelings.	[30]
*Panaeolus* *papilionaceus*	*Waraitake or odoritake* *Laughing mushroom*	(Japan)	Reports of recreative use date back to the 11th century; traditional reports indicate that *Waraitake* consumers became particularly happy, dancing, singing, and laughing compulsively, and this behavior was similar to drunk states.	[31]
*Psilocybe aztecorum*	*Apipitzin*	Nahuatls/Popocatépetl (México)	Popularly known as “rainwater child”, one of the main species used in central Mexico indigenous civilizations for entheogenic purposes.	[26]
*Psilocybe* *caerulescens* and *P. mexicana*	*Tenanácatl*	Nahuatls (Mexico)	Sacred mushroom used in religious and sacred ceremonies.	[26]
*Tricholoma* *muscarium*	No reported	No specified (Japan)	This is an important edible agaric mushroom with significant economic value in Japan. The *Tricholoma* species have an important ecological role because of the ectomycorrhizal formation with different plan families, being considered markers of conservation value measurements.	[32]

**Table 2 biomedicines-11-00461-t002:** Classification of psychoactive mushrooms, bioactive markers, and psychotropic mechanism according to Guzman et al. [42].

Group	Psychoactive Markers	Scaffold	Example	Psychotropic Mechanism	Representative Mushrooms Genus
1	Tryptamine or indole alkaloids	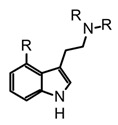	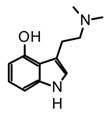 psilocin	Agonist of different serotonin receptors (5HT2a-1a-2c) and ion channels	*Psilocybe*, *Panaeolus*, *Gymnopilus*, *Copelandia*, *Agrocybe*, *Hyboloma, Galerina*, *Gerronema*, *Pluteus*, *Inocybe*, *Conocybe*, *Panaeolina*
2	Isoxazole amino acids	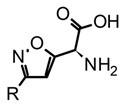	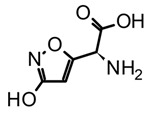 muscimol	Agonists of the ionotropic GABA_A_ receptor	*Amanita*
3	Ergot alkaloids-Indole type	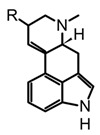	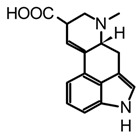 lysergic acid	Multitarget action. Partial agonist of 5-HT2, adrenergic (α1A/2), and dopamine (D2) receptors	*Claviceps* and *Cordyceps*
4	Indole-type alkaloids	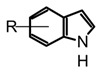 Not fully identified	Chemical studies are still required	Not elucidated yet	*Boletus*, *Heimiella*, *Russula* and some gasteromycetes

**Table 3 biomedicines-11-00461-t003:** Structure, sources, and extraction methods of some representative mushroom-derived indole alkaloids.

Indole Type	Alkaloid	Fungal Sources	Extraction Methods	Ref.
Tryptamines	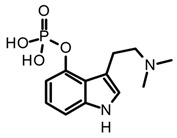 psilocybin	*Psilocybe*, *Panaeolus*, *Gymnopilus*, *Copelandia*, *Agrocybe*, *Hyboloma*, *Galerina*, *Gerronema*, *Pluteus*, *Inocybe*, *Conocybe*, *Panaeolia*.	Dry fungal biomass in the dark and at temperatures below 25 °C. Maceration or UAE with methanol or methanol with 0.5% (*v*/*v*) acetic acid at 25 °C, avoiding direct light. Homogenize samples during extraction using a vortex at 13 xG for 2 h. Re-extraction of the biomass using methanol under the conditions of temperature, agitation, and light as previously described.	[35,56,59,60,61]
	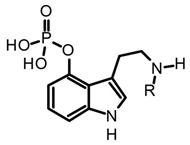 R=H norbaeocystin R=CH_3_ baeocystin	*Conocybe cyanopus*, *C. smith*, *Panaeolus cyanescens*, *Inocybe* spp., *Psilocybe* spp.		[35]
	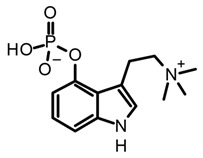 aeruginasin	*Inocybe aeruginascens*		[56,62]
	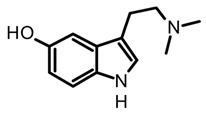 bufotenine	*Amanita citrina*, *A. porphyria*, *A. rubescens*	Dried or fresh fungal biomass. Soxhlet extraction with methanol or maceration with 50% methanol with mechanical homogenization.	[35,63,64]
β-carbolines	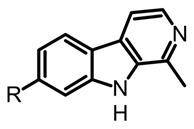 R=H harmane R=OH harmol R=OCH_3_ harmine	*Psilocybe mexicana*, *P. cyanescens*, *P. semilanceata* and *P. cubensis*	Lyophilized and grounded fungal biomass saturated with HCl 0.1 M. Extracted with dichloromethane (1:1 *v/v*). The aqueous phases basified pH 12 with NaOH and are extracted with dichloromethane.	[50]
	* 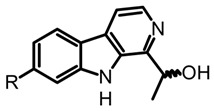 * (R)-cordysinin C (S)-cordysinin D	*Cordyceps sinensis* and *P. mexicana*		[50]
	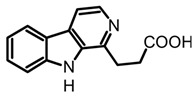 β-Carboline-1-propanoic acid	*Boletus curtisii* and *Cortinarius brunneus*	Lyophilized fruiting bodies extracted with aqueous MeOH (80%) using an ultrasound bath for 1h at room temperature. The resultant extract concentrated to dryness in a vacuum.	[55,65]
	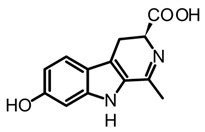 brunnein A	*Agrocybe* sp. and *C. brunneus*	Lyophilized and powdered fruiting bodies. Maceration with methanol under mechanical homogenization (1 h, 60 °C and 175 rpm).	[47,65]
Ergot	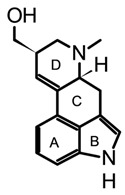 lysergol	*Claviceps* spp.	Grounded and homogenized samples. Maceration with organic basified solvents (DCM, chloroform, acetonitrile) using solvent ratios from 1:3 to 1:10 (*w*/*v*). Mechanical homogenization through vortexing or shaking (30–90 min) Organic extracts could be partitioned and pre-concentrated using solid-phase extraction (SPE).	[52,66,67]
	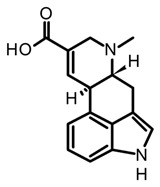 paspalic acid	*C. paspali*		
	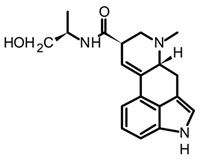 ergonovine	*C. purpurea*		

**Table 4 biomedicines-11-00461-t004:** Analytical methods for detection and quantification of psychedelic indole alkaloids in mushrooms.

Indole Type	Sample (Alkaloids Detected)	Analytical Technique	Method	Ref.
Tryptamines	*P. cubensis* and *Copelandia* spp. (Psilocin, psilocybin)	HPLC-DAD	Symmetry RP C_18_ column (150 × 2.1 mm, 5 µm). MP 10 mM ammonium formate buffer (pH 3.5) and ACN (95:5, *v*/*v*), at a flowrate of 0.2 mL/min for 20 min. Detection at 220 nm.	[61]
	*P. cubensis* (Psilocin, baeocystin, psilocybin, aeruginascin)	HPLC-ESI-MS	Zorbax Eclipse Plus RP C18 column (100 × 2.1 mm, 1.8 μm). MP 10 mmolL^−1^ ammonium formate with 0.1% (*v*/*v*) formic acid and 10 mmolL^−1^ ammonium formate with 0.1% (*v*/*v*) formic acid in methanol. Flow rate 0.25 mL min^−1^ for 7 min. For MS detection, a triple quadrupole 6460 spectrometer was used with positive ESI ionization in the dynamic multiple reaction monitoring (dMRM) acquisition mode.	[56]
	*P. mexicana* (psilocin, baeocystin, psilocybin)	HPLC-ESI-HRMS	RP C_18_ column (250 × 4.6 mm, 3 μm) with MP 0.1% C_18_ column TFA in water and ACN in gradient mode. Flow rate of 0.4 mLmin^−1^. Detection with HRMS using an exact Orbitrap spectrometer and electrospray ionization in positive mode.	[50]
	*Psilocybe spp.* and *Panaeolus cyanescens* (psilocybin)	HPLC-FL	C_18_ column (150mm × 4.6 mm, 3 μm). MP 50mM ammonium acetate (AcONH4)–CH3CN (73:27). Isocratic elution and rate flow 1.0 mLmin^−1^. FL detector at 39 nm (excitation at 321 nm).	[72]
β-carbolines	*P. mexicana* (Harmane, harmine, harmol)	HPLC-ESI-MS	RP C_18_ column (250 × 2.1 mm, 10 mm ID). MP 0.1% TFA in water and ACN. Flow 2 mLmin^−1^ with linear gradient with an increase from 10 to 100% ACN for 20 min.	[50]
	*Mycena metata* (Metatacarbolines)	HR-MALDI-MS	MALDI-MS imaging of the caps saturated solution in 80% MeOH and 1% TFA, ImagePrep device (Bruker Daltonics).	[71]
	*Cortinarius brunneus* (Brunneins)	HPLC-ESI-MS	RP C_18_ column (ODS 150 × 2.0 mm i.di, 5 μm). MP water and ACN with FA 0.2% (10:90). Using an isocratic mode for 15 min with flow 0.5 mLmin^−1^.	[65]
Ergot	*C. purpurea* (Lysergic, isolysergic, and paspalic acids)	CE-UV	P/ACE 2200 CE system with capillary (37 cm × 50 μm ID, 360 μm). The voltage applied was 25 kV, time for the separation was 12 min, and UV detection at 214 nm.	[67]
	*C. purpurea* (Ergometrine, ergotamine, ergosine, ergocryptine)	LC-MS/MS	RP C_18_ column (150 × 2.1 mm, 3.5 μm). MP water/0.2 M ammonium bicarbonate and methanol/0.2 M ammonium bicarbonate pH 10 at a flow rate of 0.15 mL/min in gradient mode. Detection with MS using triple quadrupole mass spectrometer with positive electrospray ionization.	[73]

**Table 5 biomedicines-11-00461-t005:** Preclinical pharmacological activity of natural-derived indole alkaloids related to neuropsychiatric disorders.

Type	Alkaloid	In Vitro Studies (Target Activity = _ nM)	In Vivo Studies (Preclinical Research)	Therapeutic Effects (Preclinical)	Ref.
Tryptamine	Psilocybin	h5-HT_2A_ EC_50_ = 3475 m5-HT_2B_ EC_50_ = 74 m5-HT_2C_ EC_50_ = 506 5-HT_1A_ K*i* > 10,000 5-HT_1B_ K*i* > 10,000 5-HT_1D_ K*i* = 2119 5-HT_1E_ K*i* = 194.8 5-HT_5_ K*i* = 6181 5-HT_6_ K*i* = 413.5 5-HT_7_ K*i* = 579.9	Antidepressant-like behavioral activity in chronically stressed mice with a single injection of psilocybin (1 mg/kg). Anti-compulsive-like behavior using the marble burying test in mice after treatment with a single injection of psilocybin (1 or 2 mg/kg). A single dose of psilocybin (1 mg/kg) improves stress-related behavioral deficits in mice, stimulating the growth of dendritic spines in the frontal cortex and promoting excitatory neurotransmission. A single intravenous dose of psilocybin (0.8 mg/kg) showed important changes in the pig brain, promoting synaptogenesis, increasing hippocampus density, and decreasing 5-HT_2A_R intensity. Psilocybin shown to down-regulate the functional connectivity within dopamine (DA)-associated striatal networks and increase the functional connectivity in cortical areas. A murine study showed that psilocin increases the concentrations of extracellular dopamine and serotonin in the mesoaccumbens and mesocortical pathways.	Antidepressant Anti-anhedonic Anxiolytic Anti-compulsive Cortical activation Promote synaptogenesis ↑ Neural density Potency neural circuitry ↑ Neuroplasticity	[67,80,83,84,85,86,87,88]
	Psilocin	h5-HT_2A_ EC_50_ = 4.3 m5-HT_2A_ EC_50_ = 9.9 m5-HT_2B_ EC_50_ = 58 m5-HT_2C_ EC_50_ = 30 5-HT_1A_ K*i* = 49.0 5-HT_1B_ K*i* = 219.6 5-HT_1D_ K*i* = 36.4 5-HT_1E_ K*i* = 52.2 5-HT_5_ K*i* = 83.7 5-HT_6_ K*i* = 57.0 5-HT_7_ K*i* = 3.5 α_2A_ K*i* = 1379 α_2B_ K*i* = 1894 D_3_ K*i* = 2645			
	Baeocystin/Norpsilocin	h5-HT_2A_ EC_50_ = 8.4 m5-HT_2A_ EC_50_ = 19.0 (For norpsilocin)	Mice were treated with different intravenous doses of baeocystin, and the mouse head-twitch response was measured for 20 min. No significative psychotropic-like effects were detected at doses between 0.03 to 3 mg/kg.	No reported	[68]
	Aeruginascin metabolite *	h5-HT_1D_ K*i* = 486 h5-HT_2B_ K*i* = 128 DAT K*i* = 792	No in vivo studies reported		[87]
	Bufotenine	5-HT_2A_ K*i* = 15 5-HT_2C_ K*i* = 145 5-HT_1A_ IC_50_ = 55 5-HT_1B_ IC_50_ = 29 5-HT_3_ K*i* = 34	An effective dose of 0.63 mg/day administered in mice did not cause significant physiological and behavioral effects on the animals. Bufotenine concentrations after injection (100 mg/kg) were slightly higher in the hypothalamus than in the cortex; its effects were exerted predominantly on the peripheral nervous system.		[88,89]
β-Carbolines	Harmane	MAO-A IC_50_ = 340 MAO-A K*i* = 220 MAO-B IC_50_ > 10,000 MAO-B K*i* = 57,000 5-HT_2A_ K*i* = 268 5-HT_2C_ K*i* = 2490	The systemic administration of harmane (5–20 mg/kg) in male Sprague-Dawley rats enhanced 5-HT in a dose-dependent manner. Acute intraperitoneal administration of harmane (5–20 mg/kg) in male adult rats showed antidepressant and anxiolytic effects. Intraperitoneal injection of harmane (2.5 and 10 mg/kg) in rats demonstrated potential sedative effects, also increasing corticosterone, serotonin, and noradrenaline concentrations in different regions of the brain	Antidepressant Anxiolytic	[90,91,92,93,94]
	Harmine	MAO-A IC_50_ = 8.7 MAO-A K*i* = 5.0 MAO-B IC_50_ > 10,000 5-HT_2A_ K*i* = 397 5-HT_2C_ K*i* = 5340	Mice treated with an intraperitoneal injection of harmine (20 mg/kg) for 10 days showed a significative reduction of depressive-like behaviors by ↓ of brain-derived neurotrophic factor (BDNF), ↑ the protein expression of the glutamate transporter. Male adult Wistar rats exposed to a chronic mild stress protocol were treated with harmine (15 mg/kg) for 7 days, which reversed anhedonia behavior and induced changes in the adrenal gland weight.	Antidepressant Anti-anhedonic Promote neurogenesis Stimulate neuroplasticity Repair astrocytic functions	[90,94,95,96]
	Harmaline	MAO-A IC_50_ = 11.8 MAO-A K*i* = 48 MAO-B IC_50_ > 10,000 5-HT_2A_ K*i* = 5010 5-HT_2C_ K*i* = 9430	Intraperitoneal injection of harmaline (2.5 and 5 mg/kg) produced an anxiogenic-like response in male NMRI mice, whereas 10 mg/kg of this carboline induced antidepressant-like behavior in a forced swim test.	Antidepressant Anxiolytic	[90,97]
Ergot	Ergotamine	5-HT_2A_ K*i* = 20.4 5-HT_2B_ K*i* = 6.76 5-HT_2C_ EC_50_ = 7.94 5-HT_1A_ K*i* = 12.9 5-HT_1B_ K*i* = 13.2 5-HT_1D_ K*i* = 4.36 5-HT_1E_ K*i* = 602 5-HT_1F_ K*i* = 169 5-HT_6_ K*i* = 57.0 α1_A_ K*i* = 10 α_2_ K*i* = 6.3 D2 K*i* = 3.16	Extensive studies in animals related with the vasoconstrictor effect; however, no reports related to neuropsychiatric disorders were found.	Anti-migraine	[98,99]

* The putative metabolite of aeruginascin after dephosphorylation known as 4-hydroxy-*N*,*N*,*N*-trimethyltryptamine.

**Table 6 biomedicines-11-00461-t006:** Selected examples of ongoing clinical trials involving indole alkaloids and neuropsychiatric disorders (registered on the ClinicalTrials.gov accessed on 10 October 2022).

	Intervention	Title	Conditions	Phase/ Status
Tryptamines	Psilocybin (25 mg)	The safety and efficacy of psilocybin in participants with Type 2 bipolar disorder (BP-II) Depression	Bipolar disorder, Depression	Phase 2/recruiting
	Psilocybin (10 mg in 1st session and 25 mg in 2nd session)	Psilocybin Therapy for Depression and Anxiety in Parkinson’s Disease (PDP)	Parkinson’s, Depression, Anxiety	Phase 2/recruiting
	Psilocybin (Two sessions)	Psychopharmacology of Psilocybin in Cancer Patients	Depressive symptoms Anxiety Cancer	Phase 2/ completed
	Psilocybin (0.25 mg/kg)	Efficacy of Psilocybin in OCD: a Double-Blind, Placebo-Controlled Study	Obsessive-compulsive Disorder	Phase 1/recruiting
	Psilocybin (100 or 300 µg/kg)	Psilocybin for Treatment of Obsessive-Compulsive Disorder (PSILOCD)	Obsessive-compulsive Disorder	Phase 1/in course
	Psilocybin (25 mg)	Psilocybin for Treatment-Resistant Depression	Depression	Phase 2/in course
	Psilocybin (Single dose)	The Safety and Efficacy Of Psilocybin as an Adjunctive Therapy in Participants with Treatment Resistant Depression	Resistant Depression	Phase 2/ completed
	Psilocybin (25 mg)	The Safety and Efficacy of Psilocybin in Patients with Treatment-resistant Depression and Chronic Suicidal Ideation	Resistant Depression Suicidal behavior	Phase 2/recruiting
	Psilocybin (25 mg)	A Study of Psilocybin for Major Depressive Disorder (MDD)	MDD	Phase 2/in course
	Psilocybin (0.215mg/kg)	Clinical, Neurocognitive, and Emotional Effects of Psilocybin in Depressed Patients—Proof of Concept	Depressive Disorder	Phase 2/ completed
	Psilocybin (25 mg, 2 sessions)	Psilocybin for Depression in People with Mild Cognitive Impairment or Early Alzheimer’s Disease	Depressive Symptoms Alzheimer Disease Mild Cognitive Impairment	Phase 2/recruiting
	Psilocybin (25 mg)	Psilocybin Treatment of Major Depressive Disorder with Co-occurring Alcohol Use Disorder	MDD Alcohol Use Disorder	Phase 2/recruiting
	Psilocybin (25 and 30 mg)	Psilocybin-Enhanced Psychotherapy for Methamphetamine Use Disorder	Amphetamine-Related Disorders	Phase 2/recruiting
	Psilocybin (30 mg)	Psilocybin-facilitated Smoking Cessation Treatment: A Pilot Study	Nicotine Dependence	NA
	Psilocybin (25 mg)	A Double-Blind Trial of Psilocybin-Assisted Treatment of Alcohol Dependence	Alcohol Dependence	Phase 2/ completed
	Psilocybin (25 mg)	Standardized Natural Psilocybin-assisted Psychotherapy for Tapering of Opioid Medication	Opioid Dependence Chronic Pain	Phase 2/not recruiting yet
Carboline	Harmine Harmine + DMT	Neurodynamics of Prosocial Emotional Processing Following Serotonergic Stimulation With *N*,*N*-Dimethyltryptamine (DMT) and Harmine in Healthy Subjects	Emotional and Mood disorders	Phase 2/recruiting

## Data Availability

No new data were created.

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
