# Peer review of "Indole Alkaloids from Psychoactive Mushrooms: Chemical and Pharmacological Potential as Psychotherapeutic Agents"

_biomedicines, 2023, doi:10.3390/biomedicines11020461_

Round 1
Reviewer 1 Report
The purpose of this manuscript was to evaluate the pharmacological potential of some psychoactive mushrooms along with the description of the biosynthesis methodology of some alkaloids derived from mushrooms. Due to the multiple open aspects, I believe that it is a work that will be of interest to the audience. I recommend checking the bibliographic sources and eliminating duplicate works.
Author Response
We thank the reviewer for the comments and suggestions.
The bibliographic references were double-checked, some of them have the same first author but they correspond indeed to different publications.
Reviewer 2 Report
This is a well written and organized, comprehensive review, presenting a wealth of relevant pieces of information. The focus is on history, mechanisms of action, chemistry, and analytical aspects. There is very little on the clinical data. Certainly, the paucity of the concluded trials is a reason the limited space devoted to the therapeutical findings. But, just because of it, a few sentences seem to represent an over or biased interpretation of the available data. For instance, stating that “…existing studies constitute a piece of strong evidence of their [mushroom-derived alkaloids] powerful therapeutic potential” (lines 600-601) needs to be supported, at least by a comment and by a critical description of the studies. Consistently, from a specular point of view, the statement “current pharmacotherapy for men- tal disorders, such as antidepressants, has proven to lack efficacy, reduction of sustained tolerability, and unpleasant side effects” (lines 63-65) is questionable and it requires at least a reference.
There is a very, very minor editing issue. For instance, “… indicates ultimately, the effects …) (line 53); “… known as “fly ag”aric” is one … (lines 124-125).
Author Response
We appreciate you taking the time to read and review this manuscript, as well as your suggestions and recommendations which certainly helped to improve our review.
In order to address your recommendation regarding the over or biased interpretation of the available data, the pointed-outsentences were reformulated. Also, for the sentence “Moreover, some long-term antidepressant drugs have proven…” two references were added as pieces of supporting evidence that long-term antidepressant drugs have demonstrated tolerance and side effects.
- Fornaro, M.; et al. The Emergence of Loss of Efficacy during Antidepressant Drug Treatment for Major Depressive Disorder: An Integrative Review of Evidence, Mechanisms, and Clinical Implications. Res. 2019, 139, 494–502, doi:10.1016/j.phrs.2018.10.025.
- Cartwright, C.; et al. Long-Term Antidepressant Use: Patient Perspectives of Benefits and Adverse Effects. Patient Prefer. Adherence 2016, 10, 1401–1407, doi:10.2147/PPA.S110632.
The minor editing issues were corrected in the manuscript and marked up using the “track changes” tool.
Reviewer 3 Report
The work nicely summarizes the current state of knowledge about indole alkaloids from psychoactive mushrooms. The processes of obtaining compounds are discussed in detail. The work is very interesting in terms of knowledge. The knowledge presented here may in some time be a contribution to clinical use of the above compounds. For this reason, the part describing the preclinical studies is missing the part describing the potential aspects of use. Please pay attention to important aspects such as addictive potential, suicide prevention (anti-drug and antidepressant effects), etc. I would suggest placing such a chapter before I typed conclusions.
In addition, he proposes to introduce minor corrections:
mechanisms of action in table 2 - mechanisms are described in a very brief way. What kind of modulation up, down? at what stage? receptor? conjugation with G-proteins? further feedbacks? enzyme activity? channels? additional information could be included here.
figure 2 - please enlarge the font in the legend of the figure, currently illegible
Author Response
We thank the reviewer for the comments and suggestions.
According to the reviewer’s recommendation, a new section (Safety) was added in the final part of the clinical trials to address the potential safety findings and concerns. In this new part, we described the aspects suggested such as uses related to the clinical use of natural psychedelics such as addictive potential, the use in patients with a higher risk of addictive behavior, and suicidal prevention.
In addition, minor editing issues, such as enlarging the legend in figure 2 were corrected in the manuscript. In the case of mechanisms in table 2, some additional information was provided. However, we did not go into depth, explaining all the details of the psychotropic effects of the alkaloids, since the goal of this table is to show a possible classification of psychoactive mushrooms according to their chemical composition and their possible action.
Round 2
Reviewer 3 Report
publication with corrections may be considered for publication